



**Ensemble daily simulations for elucidating cloud-aerosol interactions under**
**a large spread of realistic environmental conditions**
**Guy Dagan[1] and Philip Stier[1]**
[1] Atmospheric, Oceanic and Planetary Physics, Department of Physics, University of Oxford, UK
E-mail: guy.dagan@physics.ox.ac.uk
**Abstract**
Aerosol effects on cloud properties and the atmospheric energy and radiation budgets are
studied through ensemble simulations over two month-long periods during the NARVAL
campaigns (December 2013 and August 2016). For each day, two simulations are conducted
with low and high cloud droplet number concentrations (CDNC), representing low and high
aerosol concentrations, respectively. This large data-set, which is based on a large spread of
co-varying realistic initial conditions, enables robust identification of the effect of CDNC
changes on cloud properties. We show that increases in CDNC drive a reduction in the top of
atmosphere (TOA) net shortwave flux (more reflection) and a decrease in the lower
tropospheric stability for all cases examined, while the TOA longwave flux and the liquid and
ice water path changes are generally positive. However, changes in cloud fraction or
precipitation, that could appear significant for a given day, are not as robustly affected, and, at
least for the summer month, are not statistically distinguishable from zero. These results
highlight the need for using large statistics of initial conditions for cloud-aerosol studies for
identifying the significance of the response. In addition, we demonstrate the dependence of the
aerosol effects on the season, as it is shown that the TOA net radiative effect is doubled during
the winter month as compared to the summer month. By separating the simulations into
different dominant cloud regimes, we show that the difference between the different months
emerge due to the compensation of the longwave effect induced by an increase in ice content
as compared to the shortwave effect of the liquid clouds. The CDNC effect on the longwave is
stronger in the summer as the clouds are deeper and the atmosphere is more unstable.







## Introduction

Cloud droplets form on suitable aerosols which can serve as cloud condensation nuclei. Thus, for vertical velocities which are sufficient to sustain aerosol activation, cloud droplet number concentration (CDNC) increases with increasing aerosol concentrations. Concomitantly with the increase in the CDNC, and assuming constant liquid water content, the initial cloud hydrometeor (liquid and ice particles) size distribution shifts to smaller sizes and becomes narrower, which may modulate cloud micro- and macro-physical properties (Khain et al., 2005;Koren et al., 2005;Heikenfeld et al., 2019;Chen et al., 2017;Altaratz et al., 2014;Seifert and Beheng, 2006a;Koren et al., 2014;Dagan et al., 2017;Dagan et al., 2018b), the rain production (Levin and Cotton, 2009;Albrecht, 1989;Tao et al., 2012;Dagan et al., 2015b) and the clouds' radiative effect (Koren et al., 2010;Storelvmo et al., 2011;Twomey, 1977;Albrecht, 1989). As the anthropogenic activity involves aerosol emissions and aerosols may influence cloud radiative effects, the anthropogenic activity may perturb the Earth's radiation budget by this pathway. However, despite decades of effort of trying to better understand the processes involved, cloud-aerosol interactions are still considered one of the most uncertain anthropogenic effects on climate (Boucher et al., 2013).

The aerosol effect on clouds was previously shown to be cloud regime dependent (Altaratz et al., 2014;Lee et al., 2009;Mülmenstädt and Feingold, 2018;van den Heever et al., 2011;Rosenfeld et al., 2013;Glassmeier and Lohmann, 2016;Gryspeerdt and Stier, 2012;Christensen et al., 2016). In addition, even for a given cloud regime, small changes in the meteorological conditions may change the sign and magnitude of the aerosol effect (Dagan et al., 2015b;Fan et al., 2009;Fan et al., 2007;Kalina et al., 2014;Khain et al., 2008;Liu et al., 2019).

The fact that the aerosol effect on clouds and precipitation is dependent on the cloud regime and meteorological conditions, makes the quantification of its global effect challenging and uncertain (Mülmenstädt and Feingold, 2018;Bellouin et al., 2019). One way to overcome this challenge is by examining the aerosol effect for an ensemble of realistic co-varying initial conditions (as opposed to perturbing each environmental condition separately). This can be done by conducting ensemble/routine numerical simulations (such as those conducted in previous studies (Gustafson Jr and Vogelmann, 2015;Gustafson et al., 2017;Klocke et al., 2017)) focusing on aerosol effects. This methodology enables identifying, using large statistics, clouds and radiative properties that respond in a consistent manner to aerosol (noting that in a single-case studies some of the differences between different simulations could be just due to different realizations of the model (Grabowski, 2015)). This methodology also enables





investigation of the aerosol effect on cloud and precipitation as a function of the initial
conditions.
In a recent paper, focusing on two specific cases (each one for two days) and a relatively large
domain (22° x 11°), the physical processes controlling the aerosol effect on the atmospheric
energy budget were investigated (Dagan et al., 2019). It was shown that the total column
atmospheric radiative warming ($Q_R = (F_{SW}^{TOA} - F_{SW}^{SFC}) + (F_{LW}^{TOA} - F_{LW}^{SFC})$), defined as the rate of net
atmospheric diabatic warming due to radiative shortwave (SW) and longwave (LW) fluxes at
the surface (SFC) and top of the atmosphere (TOA), when all fluxes positive downwards), is
substantially increased with CDNC in a deep-cloud dominated case (by ~10 W/m$^2$), while a
much smaller increase (~1.6 W/m$^2$) is shown in a shallow-cloud dominated case. This trend is
caused by an increase in the upward mass flux of ice and water vapor to the upper troposphere
that leads to reduced outgoing longwave radiation. The increase in mass flux is caused partially
by an increase in vertical velocities (Koren et al., 2005;Rosenfeld et al., 2008;Dagan et al.,
2018a) and mostly by an increase in the water content at the mid-troposphere (due to warm
rain suppression) that increases the upward mass flux, even for a give vertical velocity. The
change in net radiative fluxes at the TOA ($F_{SW+LW}^{TOA}$) was shown to be -5.2 W/m$^2$ for the shallow-
cloud dominated case and -1.9 W/m$^2$ for the deep-cloud dominated case. Dagan et al. (2019)
also show that the cloud fraction responds in opposite ways to CDNC perturbations in the
different cases, increasing in the deep-cloud dominated case and decreasing in the shallow-
cloud dominated case. However, it is unclear how representative these results are as they are
based on two specific cases. The ensemble simulations presented in this study could be used to
examine the robustness of these aerosol effects using large statistics.
The focus of this study is on clouds over the Atlantic Ocean near Barbados (Fig. 1). Barbados
is located north of the mean intertropical convergence zone (ITCZ) location, in a way that
samples both the trade region, dominated by shallow cumulus during the boreal winter, and the
transition to deep convection as the ITCZ migrates northward during boreal summer (Stevens
et al., 2016). Hence, this location enables investigation of different cloud regimes and different
meteorological conditions. In addition, the clouds near Barbados have been shown to be
representative of clouds across the trade region (Medeiros and Nuijens, 2016).

**Methodology**
Ensemble daily simulations using the icosahedral nonhydrostatic (ICON) atmospheric model
(Zängl et al., 2015) in a limited area configuration are conducted. ICON's dynamical core has
been validated against several idealized cases as well as against numerical weather prediction





skill scores (Zängl et al., 2015). The domain is located east of Barbados island and covers ~3°
x 3° (Fig. 1). The simulations are aligned with the NARVAL (Next-generation Aircraft
Remote-Sensing for Validation Studies (Klepp et al., 2014;Stevens et al., 2019;Stevens et al.,
2016)) campaigns which took place during December 2013 (NARVAL 1) and August 2016
(NARVAL 2) in the northern tropical Atlantic. We use existing NARVAL convection-
permitting simulations (Klocke et al., 2017) as initial and boundary conditions for our
simulations and a two-moment bulk microphysical scheme (Seifert and Beheng, 2006b). For
each day during these two months, two different simulations are started with identical initial
conditions with different CDNC of 20 $cm^{-3}$ (clean) and 200 $cm^{-3}$ (polluted), resulting in an
ensemble of 124 simulations. The different CDNC scenarios serve as proxy for different
aerosol concentration conditions and are chosen as they represent the range typically observed
over the ocean (Rosenfeld et al., 2019;Gryspeerdt et al., 2019). Using a fixed CDNC avoid the
uncertainties involved in the representation of the aerosols processes in numerical models
(Rothenberg et al., 2018), however, it limits potential feedbacks between clouds and aerosols,
such as through involve with aerosol scavenging.
Each simulation is conducted for 24 hours starting from 12 UTC (12 hours after the original
simulations of Klocke et al., 2017 started to reduce spin-up effects). The horizontal resolution
is set to 1200 m and 75 vertical levels are used. The temporal resolution is 12 seconds and the
output interval is 30 minutes. Interactive radiation is calculated every 12 minutes using the
RRTM-G scheme (Clough et al., 2005;Iacono et al., 2008;Mlawer et al., 1997). The simulations
include an interactive surface flux scheme and a fixed (for each day) sea surface temperature.
As in Dagan et al. (2019), the simulations include representation of the Twomey effect,
calculated with diagnosed cloud droplet effective radii from the microphysical scheme
(Twomey, 1977). However, due to the large uncertainty involved in the ice microphysics and
morphology, no Twomey effect due to changes in the ice particles size distribution was
considered.
In addition, the domain is setup to include the Barbados Cloud Observatory (BCO, (Stevens et
al. 2016)) while minimising the island effect of Barbados (most of the domain is east of the
island and only the east part of the island, which includes the BCO (13°N, 59°W), is included
in the domain). Observations from the BCO are used for model evaluation (Figs. S1 and S2,
supporting information), and demonstrate that the model performs well for low surface-SW-
flux days but underestimates the flux for high-SW-flux days (usually under low cloud fraction).
We note that although a 3 ° x 3° domain is larger than the domains used in many previous
studies, it is still possible that the use of fixed boundary conditions for the different simulations



under different CDNC conditions reduces some of the sensitivity as compared to simulations
with larger domains such as in Dagan et al. (2019) (22° x 11°). Hence, the aerosol response we
present here is estimated as the lower bound.

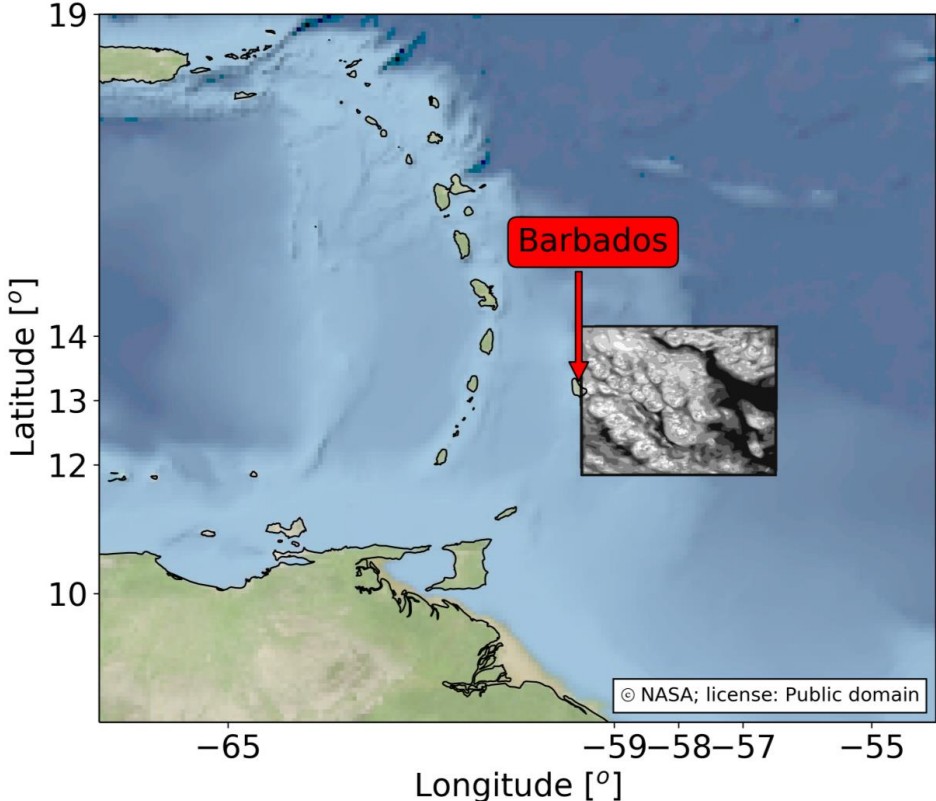


**Figure 1. The domain of the simulations (the box in the middle) and the area around it. Inside the domain**
**is presented the average cloud fraction over the first 30 mins of the simulation for 1/8/2016, CDNC = 20**
**cm⁻³. The island of Barbados is marked with a red arrow.**

**Results**
Conducting daily simulations over two months at different seasons allows us to sample a large
ensemble of initial conditions and cloud types (see Fig. 2 and Table 1). To identify statistically
significant differences between the two months, we conduct independent t-test (p-values are
presented in Table 1). This demonstrates that the lower tropospheric stability (LTS), top of
atmosphere shortwave flux ($F_{SW}^{TOA}$), and the atmospheric column radiative term ($Q_R$) are different



in a statistically significant manner (p-value < 0.05) between the two different months. The
differences in other parameters are not statistically significant (Table 1).

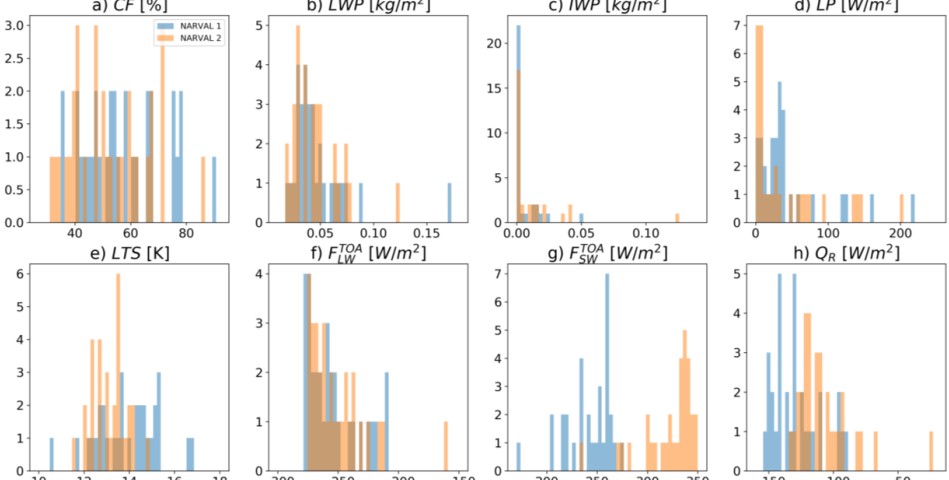

**Figure 2. Histograms of mean (time and space) cloud and atmospheric properties for the base simulations**
**with CDNC = 20 cm$^{-3}$ (clean simulations) for each day of the two months that were simulated. Blue**
**represents the NARVAL 1 month (December 2013), while orange the NARVAL 2 month (August 2016). a)**
**cloud fraction – CF, b) liquid water path - LWP, c) ice water path – IWP, d) precipitation latent heat flux**
**- LP, e) lower tropospheric stability – LTS, f) top of atmosphere longwave flux - $F_{LW}^{TOA}$ , g) top of atmosphere**
**shortwave flux - $F_{SW}^{TOA}$, and h) atmospheric column radiative term - $Q_R$.**
**Table 1. The monthly mean value of each of the properties presented in Fig. 2 ± 1 standard deviation for**
**each month and the p-value of the two-sample independent t-test. The p-values which demonstrate a**
**significant difference between the months (<0.05) are presented in bold.**

|  | Mean NARVAL 1 | Mean NARVAL 2 | p-value t-test |
|---|---|---|---|
| **CF [%]** | 57.2 ± 13.7 | 52.3 ± 13.4 | 0.16 |
| **LWP [kg/m$^2$]** | 4.8·10$^{-2}$ ± 2.8·10$^{-2}$ | 4.5·10$^{-2}$ ± 2.2·10$^{-2}$ | 0.66 |
| **IWP [kg/m$^2$]** | 5.7·10$^{-3}$ ± 1.1·10$^{-2}$ | 1.2·10$^{-2}$ ± 2.4·10$^{-2}$ | 0.19 |
| **LP [W/m$^2$]** | 43.8 ± 47.8 | 52.2 ± 78.2 | 0.6 |
| **LTS [K]** | 13.9 ± 1.4 | 13.1 ± 0.7 | **7·10$^{-3}$** |
| **$F_{LW}^{TOA}$ [W/m$^2$]** | -254.2 ± 21.2 | -251.7 ± 23.5 | 0.66 |
| **$F_{SW}^{TOA}$ [W/m$^2$]** | 241.7 ± 22.5 | 321.9 ± 26.4 | **1.4·10$^{-18}$** |
| **$Q_R$ [W/m$^2$]** | -129.2 ± 17.8 | -107.8 ± 21.7 | **9.8·10$^{-5}$** |

Figures 3 and 4 present vertical profiles of the total water (liquid and ice) mixing ratio from
the different simulations during NARVAL 2 (August 2016) and NARVAL 1 (December 2013),



respectively. Generally, during the winter month (NARVAL 1) the clouds are shallower than
in the summer month (NARVAL 2), although there is significant variability. This is expected
due to the seasonality of the ITCZ location (Stevens et al., 2016). The simulated days are
manually separated to three different cloud regimes based on the domain and time mean total
water mixing ratio vertical profiles. The cloud regimes considered here are: shallow clouds
(shallow-cloud dominated days), two-layer clouds (shallow cloud layer and a cirrus cloud
layer) and deep clouds (deep-cloud dominated days).

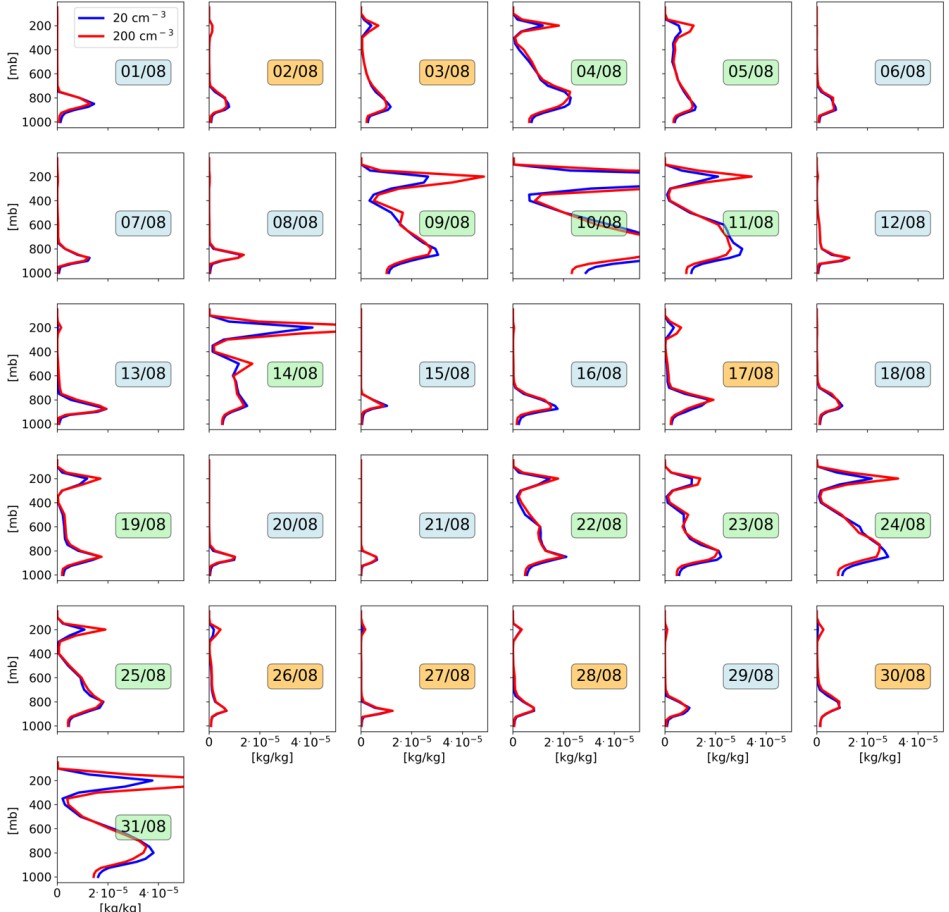

**Figure 3. Mean (time and space) vertical profiles of the total water (liquid and ice) mixing ratio in each**
**simulation (each last for 24 hours) for the NARVAL 2 month (August 2016). Blue: clean conditions (20 cm⁻**
**³), red: polluted conditions (200 cm⁻³). The simulated days are separated into three different cloud regimes:**
**shallow clouds (blue date box), two-layer clouds (shallow cloud layer and a cirrus cloud layer – orange date**
**box) and deep clouds (green date box).**






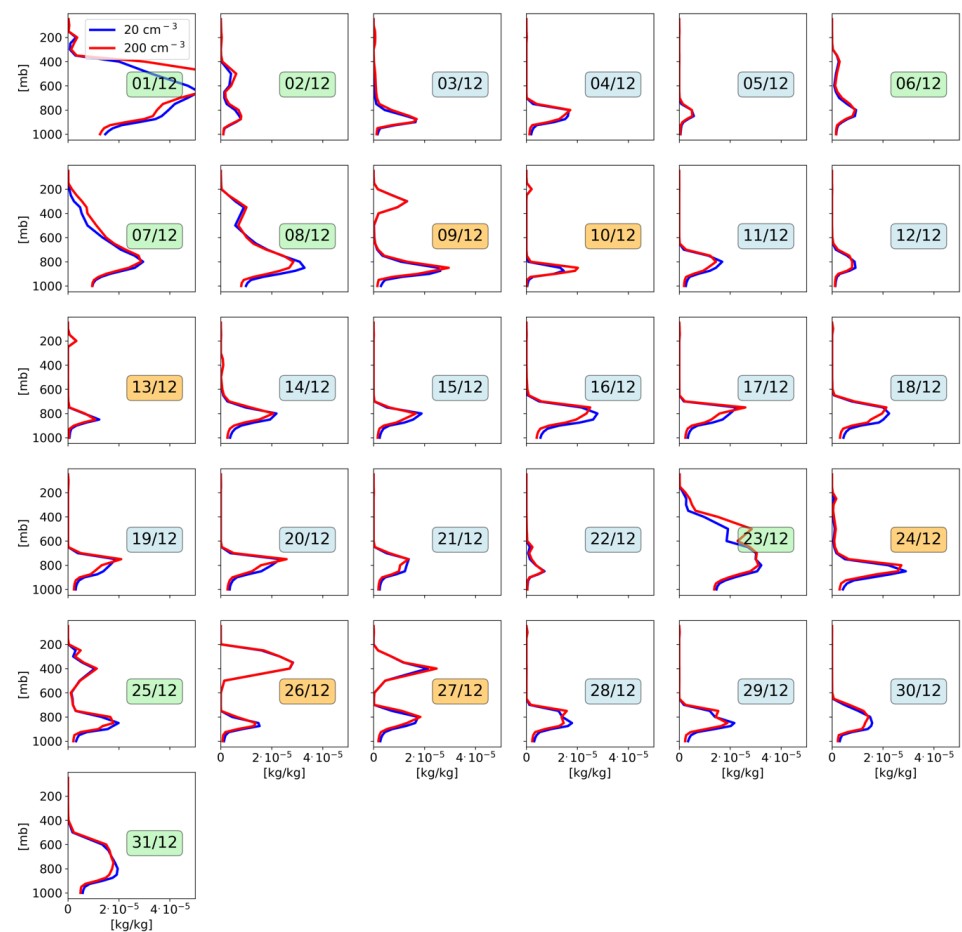


**Figure 4. Same as Fig. 3 but for the NARVAL 1 month (December 2013).**


Figure 5 presents histograms of aerosol effects (polluted minus clean) for the different
simulations. The distribution of changes in cloud fraction (Fig. 5a) demonstrate small mean
values for both months (-0.3% and 0.1% for the winter and summer month, respectively) which
is slightly more skewed to positive values in the summer. Examining the significance of these
trends with a t-test demonstrates that only the winter month response is statistically significant
(Table 2). The CDNC effect on the liquid water path (LWP; Fig. 5b) and the ice water path
(IWP; Fig. 5c) is shown to be almost entirely positive (or zero) in both months and differs from
zero in a statistically significant manner. The mean change in precipitation (Fig. 5d) is small
and negative (slightly more negative during the winter month). However, during the summer




200 month it is not statistically significant and can be either positive or negative. We note that the

201 mean precipitation decreases during the winter month (which is statistically significant) is

202 small and equivalent to 0.07 mm/day (-1.8 W/m$^2$). Increasing CDNC systematically decreases

203 LTS (Fig. 5e), representing deepening of the boundary layer (Dagan et al., 2016;Lebo and

204 Morrison, 2014;Seifert et al., 2015;Stevens and Feingold, 2009). This trend is statistically

205 significant for both months (Table 2).

206 The CDNC effect on $F_{LW}^{TOA}$ is positive and small (average of 0.24 W/m$^2$) in the winter month

207 (but still statistically significant) and larger (average of 2.16 W/m$^2$) in the summer month (Fig.

208 5f – positive flux downwards), primarily due to an increase in ice water content under polluted

209 conditions (see also Figs. 3, 4 and 5c). We previously showed that an increase in CDNC drives

210 an increase in the ice content at the upper troposphere and hence a reduction in the outgoing

211 LW radiation (Dagan et al., 2019); here we show that this trend is statistically significant (Fig.

212 5c). However, during the winter, when deep convective clouds are less abundant and the

213 atmosphere is more stable, the LW flux is less affected.

214 The CDNC effect on $F_{SW}^{TOA}$ is always negative (Fig. 5g) and is on average -3.6 W/m$^2$ and -3.8

215 W/m$^2$ in the winter and summer month, respectively (the difference between the two months

216 is not statistically significant; however, both differ from zero in a statistically significant

217 manner -Table 2). The negative $F_{SW}^{TOA}$ effect is caused mostly due to the Twomey effect

218 (Twomey, 1977) and the LWP/IWP effect (Albrecht, 1989;Koren et al., 2010;Malavelle et al.,

219 2017) (Figs. 5b and 5c), as the CF changes are small (Fig. 5a). For exploring the relative role

220 of the Twomey and IWP/LWP effects, we ran all simulations again with the Twomey effect

221 turned off.  Without the Twomey effect the SW effect is reduced by up to a factor of 10 (-0.35

222 W/m$^2$ compared with -3.6 W/m$^2$ in the winter month, and -1.0 W/m$^2$ compared with -3.8 W/m$^2$

223 in the summer month). This demonstrates that the Twomey effect is the dominant factor

224 underlying the $F_{SW}^{TOA}$ changes. Radiative effects due to changes in ice size distribution are not

225 considered due to uncertainties in the evolution of ice morphology. Accounting for this effect

226 would further increase the relative role of the Twomey effect compare to the cloud adjustment

227 effects (CF and LWP/IWP adjustments).

228 The change in the atmospheric column radiative warming term $Q_R$ is shown to be small for the

229 winter month (-0.26 W/m$^2$ on average) but much larger and positive for the summer month

230 (1.8 W/m$^2$ on average). The increase in $Q_R$ during the summer is caused due to the effect of

231 deep, ice containing clouds on the outgoing LW flux (Fig. 5f). SW flux changes due to CDNC

232 perturbations (Fig. 5g) have a much smaller effect on $Q_R$ as the SW absorption of clouds is

233 small (Dagan et al., 2019).





Examining the similarity between the response of the different properties to the CDNC
perturbation in the two different months (Table 2) reveals that the responses of the IWP, $F_{LW}^{TOA}$,
$Q_R$ and $F_{SW+LW}^{TOA}$ (the net TOA LW and SW effects – Fig. 10 below) are different in a statistically
significant manner between the two months. As will be shown below, this is related to the
response of the ice content.


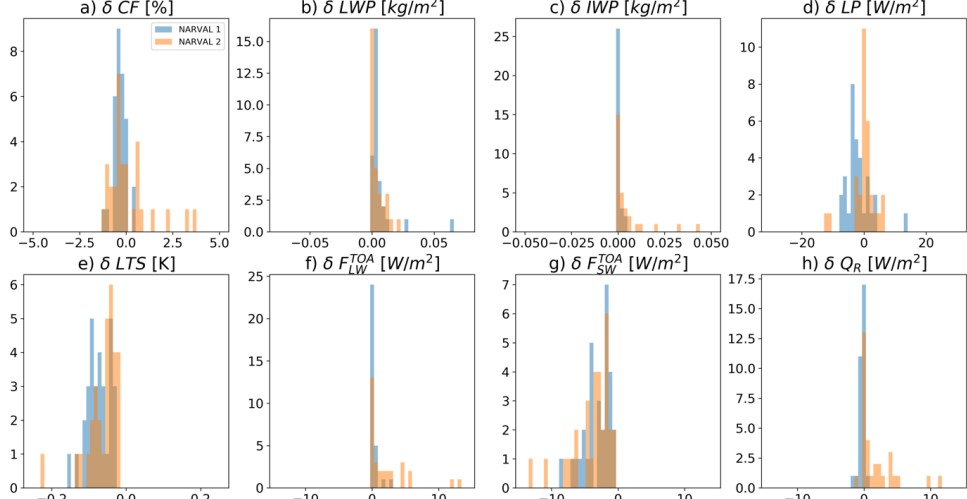


**Figure 5. Histograms of the domain and time mean response of cloud and atmospheric properties to CDNC**
**perturbation (polluted simulations minus clean simulations) for each day of the two months that were**
**simulated. Blue represents the NARVAL 1 month (December 2013), while orange the NARVAL 2 month**
**(August 2016). a) cloud fraction – CF, b) liquid water path - LWP, c) ice water path – IWP, d) precipitation**
**latent heat flux - LP, e) lower tropospheric stability – LTS, f) top of atmosphere longwave flux - $F_{LW}^{TOA}$, g)**
**top of atmosphere shortwave flux - $F_{SW}^{TOA}$, and h) atmospheric column radiative term - $Q_R$.**













**Table 2. Summary of monthly mean response of cloud and atmospheric properties (presented in Fig. 5) to**
**the CDNC perturbation (polluted simulations minus clean simulations) ± 1 standard deviation for each**
**month. In addition, the p-values of the two-sample independent t-test are presented, as well as the p-values**
**for comparing the CDNC response in each month to zero. The p-values which demonstrate significant**
**difference (<0.05) are presented in bold.**

| | Mean NARVAL 1 | Mean NARVAL 2 | p-value t-test | p-value one sample t-test compare to 0 - NARVAL 1 | p-value one sample t-test compare to 0 - NARVAL 2 |
|---|---|---|---|---|---|
| $\delta$CF [%] | $-0.32 \pm 0.31$ | $0.11 \pm 1.15$ | 0.053 | **$8.1\cdot10^{-6}$** | 0.6 |
| $\delta$LWP [kg/m²] | $6.5\cdot10^{-3} \pm 1.2\cdot10^{-2}$ | $4.0\cdot10^{-3} \pm 5.4\cdot10^{-3}$ | 0.3 | **$4.4\cdot10^{-3}$** | **$3.5\cdot10^{-4}$** |
| $\delta$IWP [kg/m²] | $5.6\cdot10^{-4} \pm 1.3\cdot10^{-3}$ | $8.2\cdot10^{-3} \pm 1.9\cdot10^{-2}$ | **0.035** | **0.02** | **0.03** |
| $\delta$LP [W/m²] | $-1.8 \pm 4.1$ | $-1.2 \pm 7.0$ | 0.7 | **0.02** | 0.37 |
| $\delta$LTS [K] | $-0.075 \pm 0.031$ | $-0.062 \pm 0.042$ | 0.18 | **$3.2\cdot10^{-14}$** | **$4.3\cdot10^{-9}$** |
| $\delta F_{LW}^{TOA}$ [W/m²] | $0.24 \pm 0.60$ | $2.16 \pm 3.25$ | **0.002** | **0.03** | **0.001** |
| $\delta F_{SW}^{TOA}$ [W/m²] | $-3.6 \pm 3.5$ | $-3.8 \pm 2.9$ | 0.8 | **$3.3\cdot10^{-6}$** | **$4.7\cdot10^{-8}$** |
| $\delta Q_R$ [W/m²] | $-0.26 \pm 0.39$ | $1.8 \pm 2.8$ | **$1.8\cdot10^{-4}$** | **$9.7\cdot10^{-4}$** | **$1.4\cdot10^{-3}$** |
| $\delta F_{SW+LW}^{TOA}$ | $-3.36 \pm 3.02$ | $-1.67 \pm 1.93$ | **0.01** | **$1.1\cdot10^{-6}$** | **$5.1\cdot10^{-5}$** |


## CDNC effect on different cloud regimes

For better understanding the trend demonstrated in Fig. 5 and Table 2, we split the simulated
days into different dominant cloud types/regimes (see Figs. 3 and 4). Figures 6 and 7 present
histograms of the same atmospheric properties presented in Fig. 2 but separated by different
cloud regimes – shallow clouds, two-layer clouds (shallow clouds with cirrus cloud layer
above), and deep clouds. These figures demonstrate that the cloud fraction, LWP, IWP,
precipitation, $F_{LW}^{TOA}$ and $Q_R$ are generally higher on days dominated by deep-clouds as compared
to days dominated by shallow clouds, while the LTS and $F_{SW}^{TOA}$ are lower in the deep-cloud
dominated days compared to shallow-cloud dominated days (with the two-layer cloud days
generally in-between them). The separation into different cloud regimes also demonstrates that
more deep-cloud days are occurring during the summer month as compared to the winter month
(12 compare to 8) and that the deep clouds during summer are deeper and contain more water.




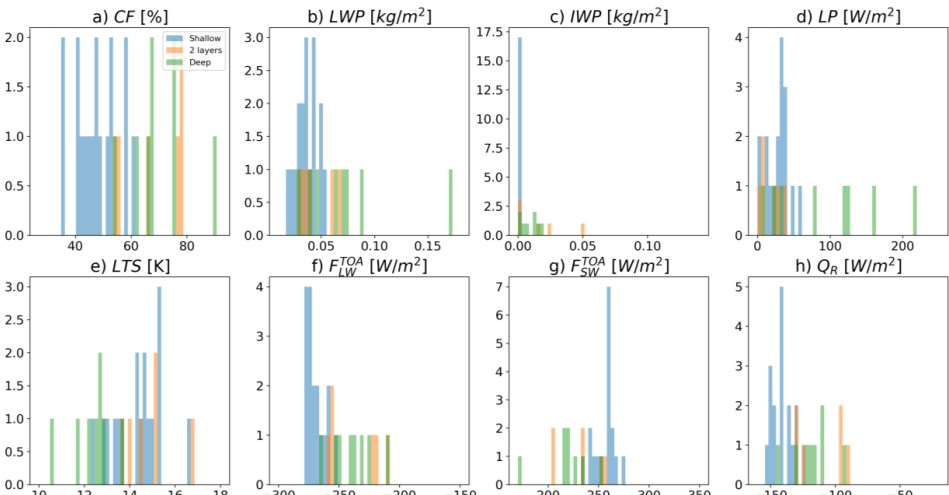

**Figure 6. Histograms of mean (time and space) cloud and atmospheric properties for the base simulations with CDNC = 20 cm$^{-3}$ (clean simulations) for each day of the NARVAL 1 month (December 2013) separated into different cloud regimes: shallow clouds (blue), two-layer clouds (shallow clouds with cirrus clouds layer above - orange), and deep clouds (green). a) cloud fraction – CF, b) liquid water path - LWP, c) ice water path – IWP, d) precipitation latent heat flux - LP, e) lower tropospheric stability – LTS, f) top of atmosphere longwave flux - $F_{LW}^{TOA}$, g) top of atmosphere shortwave flux - $F_{SW}^{TOA}$, and h) atmospheric column radiative term - $Q_R$.**

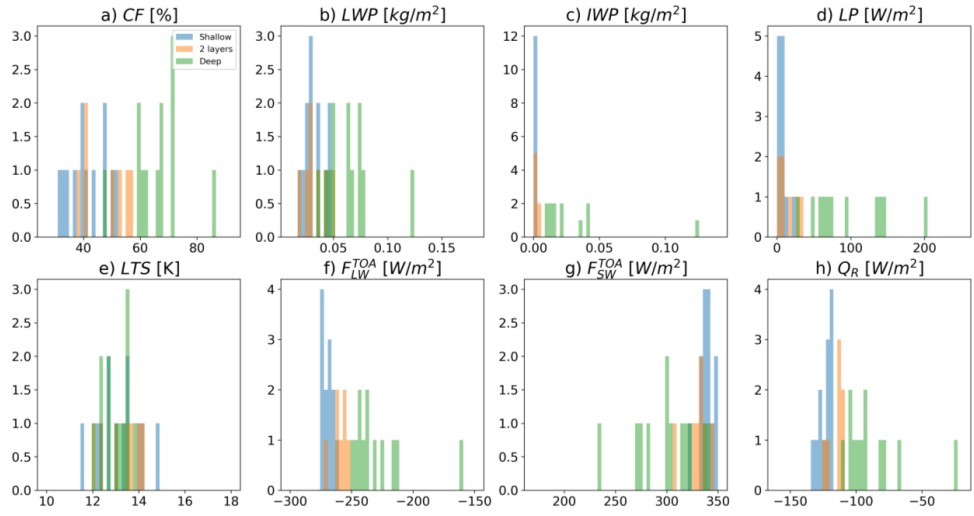

**Figure 7. Same as Fig. 6 but for the NARVAL 2 month (August 2016).**



Examining the response of the different cloud regimes to the CDNC perturbation (Figs. 8 and
9) demonstrates that the response of the cloud fraction, LWP, IWP and $F_{LW}^{TOA}$ in the deep-cloud
days is generally more positive, while the response of $F_{LW}^{TOA}$ and LTS is generally more negative.
These trends are more pronounced during the summer month as compared to the winter month.
The response of $Q_R$ is more positive in the deep-cloud dominated days in the summer month
but does not show any different trend in the winter month. The precipitation response does not
show any distinct different trend for the different cloud types in both months.
The findings presented in Figs. 8 and 9 demonstrate that the IWP response in the deep-cloud
dominated days is generally stronger in the summer month as compare to the winter month.
The increase in the IWP with the increase in CDNC drives a reduction in $F_{LW}^{TOA}$ and hence
increase in $Q_R$ (Dagan et al., 2019). We note that the largest difference between the two months
emerges due to the stronger response of the ice content in the summer month as compared to
the winter month. This fact can explain the statistically significant different response of the
IWP, $F_{LW}^{TOA}$ and $Q_R$ shown in Table 2.

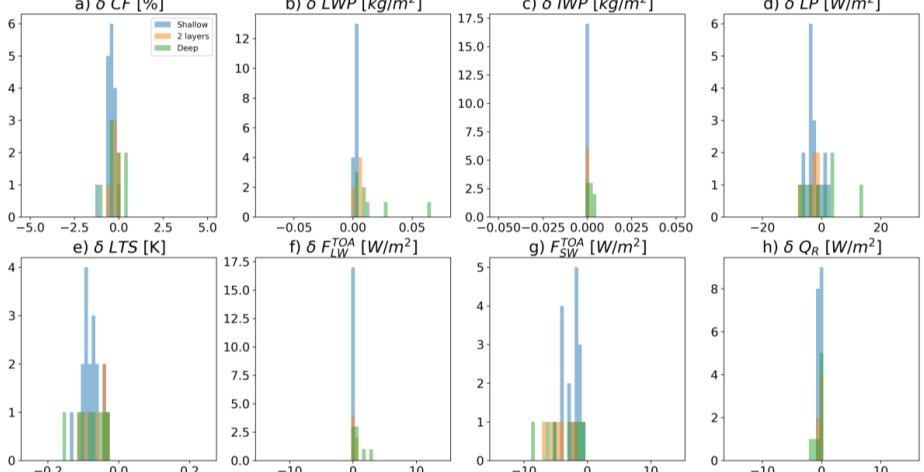

**Figure 8. Histograms of the domain and time mean response of cloud and atmospheric properties to the**
**CDNC perturbation (polluted simulations minus clean simulations) for each day of the NARVAL 1 month**
**(December 2013) separated into the different cloud regimes: shallow clouds (blue), two-layer clouds**
**(shallow clouds with cirrus clouds layer above - orange), and deep clouds (green). a) cloud fraction – CF,**
**b) liquid water path - LWP, c) ice water path – IWP, d) precipitation latent heat flux - LP, e) lower**
**tropospheric stability – LTS, f) top of atmosphere longwave flux - $F_{LW}^{TOA}$ , g) top of atmosphere shortwave**
**flux - $F_{SW}^{TOA}$, and h) atmospheric column radiative term - $Q_R$.**






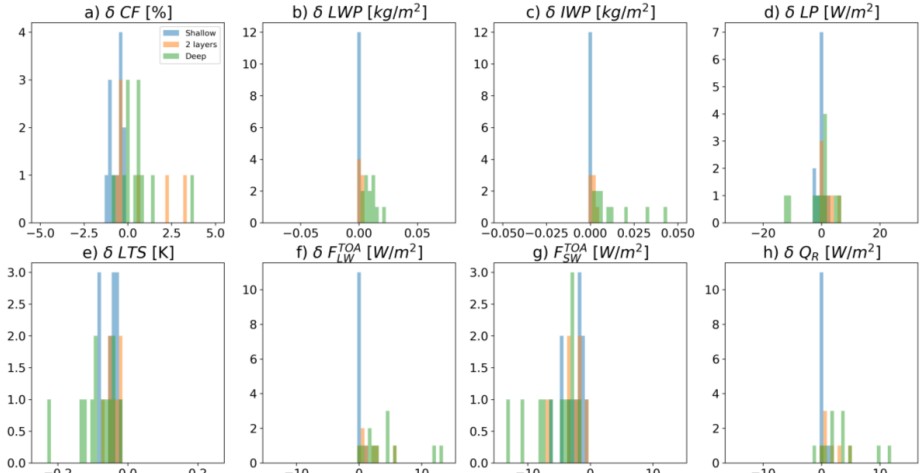

**Figure 9. Same as Fig. 8 but for the NARVAL 2 month (August 2016).**

The combined CDNC effect on the total net TOA radiation ($F_{SW+LW}^{TOA}$) is shown in Fig. 10. It
demonstrates that during the winter month the effect on $F_{SW+LW}^{TOA}$ is always negative and has a
mean value of -3.4 W/m$^2$. However, during the summer month, the mean effect is less negative
(-1.7 W/m$^2$) and for some of the days it could even be positive due to the effect of the CDNC
on the ice water content (Fig. 5 and Table 2). The difference between the two months in $F_{SW+LW}^{TOA}$
is statistically significant (Table 2). We note that during the summer month all days for which
$F_{SW+LW}^{TOA} \geq 0$ are deep-cloud dominated days, supporting the hypothesis that the difference
between the different months are driven by the different response of the deep clouds, which are
deeper and contain more water in the summer month.

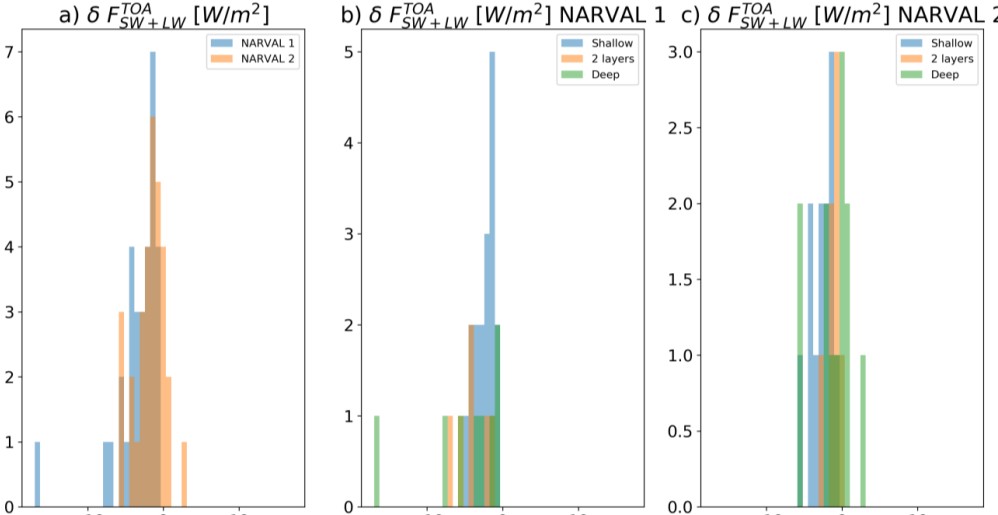

**Figure 10. Histograms of the response of the net (shortwave + longwave) top of atmosphere radiative flux ($F_{SW+LW}^{TOA}$) to the CDNC perturbation (polluted simulations minus clean simulations) for each of the simulated days. In a) blue represents the NARVAL 1 month (December 2013), while orange the NARVAL 2 month (August 2016). In b) and c) the NARVAL 1 and the NARVAL 2 months are separated to the different cloud regimes: shallow clouds (blue), two-layer clouds (shallow clouds with cirrus clouds layer above - orange), and deep clouds (green).**

## Summary

Ensemble daily simulations over a region near Barbados for two separate month-long periods were conducted to investigate aerosol effects on cloud properties and the atmospheric energy budget. For each day, two simulations were conducted with low and high CDNC representing clean and polluted conditions, respectively. These simulations are used to distinguish between properties that are robustly affected by changes in CDNC and those that are not. For example, we have shown that, for the entire set of simulations (62 different days), an increase in CDNC always drives a reduction in the lower tropospheric stability (Fig. 5). In addition, $F_{SW}^{TOA}$ is always reduced by an increase in CDNC, representing more SW reflection. However, changes in cloud fraction or precipitation are not as robustly affected, and, despite the fact that for a given day they could be large, on average they are not distinguishable from zero (at least for the summer month). However, we note that the aerosol response we present here may be underestimate due to the effect of the fixed boundary conditions and hence is estimated as the lower bound.

In addition, the use of two month-long periods, covering different seasons dominated by different meteorological conditions and cloud type, demonstrate again (Altaratz et al.,



2014;Lee et al., 2009;Mülmenstädt and Feingold, 2018;van den Heever et al., 2011;Rosenfeld
et al., 2013;Glassmeier and Lohmann, 2016;Gryspeerdt and Stier, 2012;Dagan et al., 2015a),
that the aerosol effect on clouds is strongly dependent on cloud regimes and meteorological
conditions. For our simulations we demonstrate that the top of atmosphere net radiative effect
is twice as large during the winter month as compared to the summer month (Fig. 10).
To better understand these differences we have split the simulated days into three different
dominant cloud regimes. The results demonstrate that most of the differences in the response
to CDNC increases between the two months are driven by the response of the ice content in
deep convective clouds. During the summer month, the atmosphere is less stable and the deep
convective clouds in the base-line simulations are more abundant, reach higher levels in the
atmosphere and contain more water. These more developed clouds respond stronger to the
CDNC perturbations and develop more ice content than the shallower clouds during the winter
month. The increased ice is driven by increase in mass flux to the upper levels. The added ice
content reduces the outgoing LW flux at the TOA and hence compensates some of the SW
effect, which itself is similar between the summer and winter months.
Our results highlight the need to use large ensembles of initial conditions for cloud-aerosol
interaction studies, even in large domain simulations, and suggest that caution is needed when
trying to draw conclusions from a single case-study experiments and short-term observations.

**Author contributions.** G. D. carried out the simulations and analyses presented. P. S. assisted
with the design and interpretation of the analyses. G.D. prepared the manuscript with
contributions from P.S.

**Acknowledgements:**
This research was supported by the European Research Council (ERC) project constRaining
the EffeCts of Aerosols on Precipitation (RECAP) under the European Union's Horizon 2020
research and innovation programme with grant agreement No 724602. The simulations were
performed using the ARCHER UK National Supercomputing Service. We acknowledge MPI,
DWD and DKRZ for the NARVAL simulations.

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
