# Peer review of "Supporting information of"

_Atmospheric Chemistry and Physics, 2019_

## Referee Comment (RC1) · Anonymous Referee #1 · 6 Jan 2020

This study performs a sensitivity test to the CCN concentrations in a domain of 3x3 degrees just to the west of Barbados. It runs two full months of actual weather, one in December 2013 and one in August 2016. A major conclusion that is well supported by the study is the variability of the indicated aerosol effects on different days, and the implication that conclusions from single case studies should not be generalized for a large range if situations.

The rest of the quantitative conclusions of the study with respect to aerosol effects are limited by the fidelity of the model that was used and to the way of its application. The

model that was used is a two-moment bulk microphysical scheme (Seifert and Beheng, 2006b). The model has severe major limitations:

1. The model assumes saturation adjustment, thus cannot realize the invigoration mechanism that is incurred by the warm cloud invigoration mechanism mediated by the aerosol control on the supersaturation that was co-authored by the first author of this study (Koren et al., 2014). In that paper it is shown that the lack of supersaturation adjustment is responsible for most of the substantial invigoration of water convective clouds on the background of very low CCN.

2. The mechanism of convective invigoration mediated by aerosol control of the super-saturation becomes even more important in deep convective clouds (Fan et al., 2018). Therefore, selecting a model with saturation adjustment misses most of the aerosol effect on deep convective clouds.

3. Furthermore, in the model, droplet nucleation does not change the CCN spectrum, as acknowledged in the manuscript. But the scavenging of aerosol by precipitation serves as a strong positive feedback to amplify the difference in the aerosol effect between raining and not raining clouds.

4. In addition, the 2M scheme does not suppress rain in high CCN concentrations to the extent that occurs in reality, where rain is suppressed pretty much when cloud drop effective radius is smaller than 14 micrometer (Chen et al., 2008; Freud et al., 2012; Gerber, 1996; Prabha et al., 2011; Rosenfeld et al., 2012; Van Zanten et al., 2005).

5. The model misses the processes which can lead to positive net TOA warming due to aerosols, as simulated using SBM by Fan et al. (2012).

6. The model resolution of 1200 m is insufficient to resolve properly the trade wind cumulus.

7. The simulation is allowed 12 hours for spin up time. This also means that clouds in air mass that enter the border of the domain (typically from the east) require that much

time to spin up. But the whole domain of 3 degrees from east to west is 325 km, divided by 12 hours equals 27 km/hour or 7.5 m/s. This means that when air mass speed is larger than that, the spin up would not be reached throughout the domain. The actual mean surface wind at Barbados airport is easterly 14 knots, which is 7.2 m/s, with little variation between winter and summer. Therefore, most of the simulated clouds are well within the spin-up time.

All these problems lend very little credibility to the conclusions with respect to the quantitative aerosol effects on the clouds. Presently SBM simulations are possible for the domain of this study, although quite more expensive. The fact that bulk models run faster is no longer a justification to use them for evaluating aerosol microphysical effects without addressing these issues.

In summary, I would recommend publication only after all these caveats will be explicitly highlighted, and the conclusions of the paper will clearly take them into account.

What is the point to run very fast with 2-Moment bulk scheme when running in the wrong direction?

References:

Chen R., R. Wood, Z. Li, R. Ferraro, F.-L. Chang, Studying the vertical variation of cloud droplet effective radius using ship and space-borne remote sensing data. J. Geophys. Res. 113, D00A02 (2008). doi: 10.1029/2007JD009596

Fan, J., D. Rosenfeld, Y. Ding, L. R. Leung, and Z. Li, 2012: Potential aerosol indirect effects on atmospheric circulation and radiative forcing through deep convection. Geophys. Res. Lett., doi:10.1029/2012GL051851.

Fan, J., Rosenfeld, D., Zhang, Y., Giangrande, S.E., Li, Z., Machado, L.A., Martin, S.T., Yang, Y., Wang, J., Artaxo, P. and Barbosa, H.M., 2018. Substantial convection and precipitation enhancements by ultrafine aerosol particles. Science, 359(6374), pp.411-418. http://dx.doi.org/10.1126/science.aan8461.

Freud E., and D. Rosenfeld, 2012: Linear relation between convective cloud drop number concentration and depth for rain initiation. J. Geophys. Res., 117, D02207, doi:10.1029/2011JD016457.

Gerber, H. Microphysics of Marine Stratocumulus Clouds with Two Drizzle Modes. J. Atmos. Sci. 53, 1649–1662 (1996).

Koren I., G. Dagan, O. Altaratz, From aerosol-limited to invigoration of warm convective clouds. Science 344, 1143–1146 (2014).

Prabha T. V. et al., Microphysics of Premonsoon and Monsoon Clouds as Seen from In Situ Measurements during the Cloud Aerosol Interaction and Precipitation Enhancement Experiment (CAIPEEX). J. Atmos. Sci. 68, 1882–1901 (2011). doi: 10.1175/2011JAS3707.1

Rosenfeld D., H. Wang, P. J. Rasch, The roles of cloud drop effective radius and LWP in determining rain properties in marine stratocumulus. Geophys. Res. Lett. 39, L13801 (2012). doi: 10.1029/2012GL052028.

vanZanten M. C., B. Stevens, G. Vali, D. H. Lenschow, Observations of Drizzle in Nocturnal Marine Stratocumulus. J. Atmos. Sci. 62, 88–106 (2005). doi:10.1175/JAS-3355.1

---

## Referee Comment (RC2) · Michael Diamond (Referee) · 12 Mar 2020

This study is a follow-up to a paper exploring two case studies from the NARVAL campaign that uses ensemble simulations of two months (of which the original cases were a subset) to analyze the robustness of inferences regarding aerosol-cloud interactions that can be made on the basis of a small number of cases. Certain changes (such as in shortwave reflections and boundary layer deepening that lowers lower tropospheric stability) appear robust whereas others (such as cloud fraction and precipitation changes) appear less so. Seasonal differences in response can be explained via

different responses in different cloud regimes, particularly due to ice-phase effects in deep clouds during the summer.

The manuscript is in very good shape and only requires some very minor revisions, in my estimation. If not for the comment below regarding the reasonableness of the "lower bound" language, I'd be happy to accept as is.

Specific comments:

Page 2, Line 45: "As the anthropogenic activity..." is phrased somewhat awkwardly. Perhaps you can simplify to something like "Anthropogenic aerosol emissions may thus perturb Earth's radiation budget both directly by scattering and absorbing light and also indirectly through these cloud-mediated mechanisms."

Page 4, Line 116: I'm glad you address this point. However, did you mean "interactions" or "feedbacks" instead of "involve"? Also, a relevant citation for the aerosol scavenging idea:

Yamaguchi, T., Feingold, G., & Kazil, J. (2017). Stratocumulus to Cumulus Transition by Drizzle. Journal of Advances in Modeling Earth Systems, 9(6), 2333-2349. doi:10.1002/2017MS001104

Page 5, Line 138: I'm not convinced this is a reasonable lower bound, given that the relatively small domain size with fixed boundary conditions (which you argue would lead to an underestimate of aerosol effects) is not the only potential source of error, or necessarily the largest. I'd either like to see a fuller explanation of why the estimates should be seen as true lower bounds or a weaker statement simply explaining this particular source of error would tend to underestimate the effect compared to a simulation with a larger domain.

Page 6, Figure 2: It would be helpful if "LP" were defined somewhere in the text in addition to in the figure captions.

Page 9, Line 226: I would add "likely" between "would" and "further" given that icephase microphysical changes can be quite complex.

Page 11, Line 275: How significant is 12 versus 8 in this context? Is there any way to quantify the variability we could expect in deep-cloud days due to chance?

References: There are some typos and weird formatting issues with some references. A quick proofread should sort most of those out.
* * *

---

## Author Comment (AC1) · 28 Mar 2020

**Ensemble daily simulations for elucidating cloud-aerosol interactions under** a large spread of realistic environmental conditions**

We would like to thank the reviewers for their constructive and thoughtful comments that helped us to improve our paper.

Please find a point by point reply to all of the reviewers' comments (in blue) below.

**Reviewer #1:**

This study performs a sensitivity test to the CCN concentrations in a domain of 3x3 degrees just to the west of Barbados. It runs two full months of actual weather, one in December 2013 and one in August 2016. A major conclusion that is well supported by the study is the variability of the indicated aerosol effects on different days, and the implication that conclusions from single case studies should not be generalized for a large range if situations. The rest of the quantitative conclusions of the study with respect to aerosol effects are limited by the fidelity of the model that was used and to the way of its application. The model that was

by the fidelity of the model that was used and to the way of its application. The model that was used is a two-moment bulk microphysical scheme (Seifert and Beheng, 2006b). The model has severe major limitations:

Reply: Before replying to each point, we would like to start with a general remark. The main concern of the reviewer is the use of a two-moment bulk microphysical scheme as opposed to a bin scheme. We agree with the reviewer that bin schemes can have advantages in representing some of the aerosol effect on clouds, and as the reviewer mentions below, the first author of this study worked quite a bit with that type of microphysical schemes previously (i.e. (Dagan et al., 2015a;Dagan et al., 2015b, 2018;Dagan et al., 2017;Dagan et al., 2016;Heiblum et al., 2016). Hence, we fully appreciate and are aware of the limitations of our model. However, one should also remember that bin microphysical schemes have their own limitations such as broadening of the droplet size distribution due to numerical diffusion (Morrison et al., 2018). In addition, different bin microphysical schemes demonstrate a high inter-model spread in predicting warm rain production (VanZanten et al., 2011, Adrian Hill, personal communication) as well as a large spread in predicting ice processes (Grabowski et al., 2019; Xue et al., 2017). This large uncertainty in microphysics schemes – of all kinds – is also supported by ongoing work in the deep convection case study of the Aerosol, Clouds, Precipitation and Climate (ACPC) initiative (van den Heever et al., in prep.). Overall, limited

observations leave the representation of mixed-phase cloud microphysics poorly constrained. More fundamentally, it is not a given (and often not true) that more complex model representations, such as bin microphysics schemes in this case, will by default evaluate better against observations than simpler representations. Instead, we would argue that the field has not yet reached a consensus about the optimal representation of cloud microphysics (Grabowski et al., 2019) and similar arguments as made in this review could be about the use of bin schemes versus fully lagrangian microphysics or superdroplet schemes.

In addition to the ongoing debate in our field about bin versus bulk microphysical approaches, there is the clear issue of computational expense which limits many bin-model studies to small domains, coarser resolution or shorter simulations (Khain et al., 2015). Addressing the representativeness of such simulations is a particular concern of our work: the main aim of our current study is to examine aerosol effect under a *wide range of initial conditions*, which requires numerus simulations. To do so we simulate 248 day-long simulations (62 days times two CDNC conditions per day times two simulations for each case to estimate the Twomey effect). Our simulations are conducted on relatively large domain of ~300 x 300 km (compared with many previous studies) with relatively high vertical and horizontal resolutions (75 vertical levels and 1.2 km horizontal resolution). This large amount of simulations required a huge amount of computational time. Using a bin microphysical scheme would enlarge the computation time by 1-2 orders of magnitude. Hence, this is impossible for us to conduct at this stage and we do not believe that it would affect our main conclusions.

For studying cloud-aerosol interactions using numerical simulation one must make some simulation choices. Currently, it is impossible to simulate cloud-aerosol interactions with the most detailed microphysical scheme (if we knew the optimal approach), with the highest spatial and temporal resolution, a large domain and a large ensemble of initial conditions. We must make a choice. In this study, we choose to focus on the effect of the initial conditions and examine a wide range of them (which is the novelty of this study). Other studies, e.g. the ACPC deep convection case study, will focus on convective cloud microphysics. As will be explained in more detail below, even though we are aware of the limitation of our simulation choices, we still think (and the reviewer seems to agree) that the main conclusion of our paper (citing from the reviewer above "…the variability of the indicated aerosol effects on different days, and the implication that conclusions from single case studies should not be generalized for a large range if situations") is "well supported".

1. The model assumes saturation adjustment, thus cannot realize the invigoration mechanism that is incurred by the warm cloud invigoration mechanism mediated by the aerosol control on the supersaturation that was co-authored by the first author of this study (Koren et al., 2014). In that paper it is shown that the lack of supersaturation adjustment is responsible for most of the substantial invigoration of water convective clouds on the background of very low CCN.

Reply: Thank you for this comment. As the reviewer mentioned we are well aware of the limitation of using saturation adjustment and worked quite a bit on this topic previously. However, please note that in our simulations we use a temporal resolution of 12 sec. The phase change relaxation time of condensation and evaporation is usually on the order of a few seconds, and even under extremally clean conditions is not more than 10 sec (Pinsky et al., 2013). Hence, even if we would use a microphysical scheme that explicitly resolves condensation and evaporation, the humidity is expected to get back to saturation on shorter time scales then the temporal resolution of the model, and hence, practically we will be in "saturation adjustment" conditions anyway.

Following this comment, we added clarifications to the revised manuscript:

"In addition, we note that use of a microphysical scheme which assumes saturation adjustment reduces the sensitivity of the clouds to some of the aerosol effect (Koren et al., 2014; Dagan et al., 2015a; Heiblum et al., 2016; Fan et al., 2018)."

"In addition, using a microphysical scheme that assumes saturation adjustment reduces the sensitivity of the clouds to aerosol perturbation (Koren et al., 2014; Dagan et al., 2015a; Heiblum et al., 2016; Fan et al., 2018). However, this might be a small effect in our case as the phase change relaxation time of condensation and evaporation is usually on the order of a few seconds (Pinsky et al., 2013). Hence, even if we would use a microphysical scheme that explicitly resolves condensation and evaporation, the humidity is expected to get back to saturation on shorter time scales then the temporal resolution of the model (12 sec), and hence, practically we will be in "saturation adjustment" conditions anyway."

2. The mechanism of convective invigoration mediated by aerosol control of the supersaturation becomes even more important in deep convective clouds (Fan et al., 2018). Therefore, selecting a model with saturation adjustment misses most of the aerosol effect on deep convective clouds.

Reply: As was mentioned above, one must make some simulation choices and, in this case, in order to sample a large spread of initial conditions, we choose to use a temporal resolution

which will anyway simulate conditions close to saturation adjustment. Other studies will rightfully focus on the representation of microphysics. In addition, whether or not saturation adjustment is responsible for *most* of the aerosol effect on deep convective clouds is still to be determined.

In the revised manuscript we acknowledge that our simulations do not include the abovementioned effect and we cite the paper the reviewer mentioned (see above).

3. Furthermore, in the model, droplet nucleation does not change the CCN spectrum, as acknowledged in the manuscript. But the scavenging of aerosol by precipitation serves as a strong positive feedback to amplify the difference in the aerosol effect between raining and not raining clouds.

Reply: Thank you for this comment. The scavenging of aerosol by precipitation could also serve as a "buffering" mechanism and reduce the difference between clean and polluted conditions. For example, if due to cloud invigoration mechanisms, initially there would be stronger rain rates under polluted conditions (Fan et al., 2013), and hence more scavenging, this could clean the atmosphere and bring it closer to the low aerosol case. However, in marine stratocumulus clouds it was shown, as the reviewer mentioned, that scavenging could amplify the difference between clean precipitating conditions and polluted non-precipitating conditions (Koren and Feingold, 2011).

The lack of inclusion of the scavenging mechanism and the feedback involved is mentioned in the manuscript:

"Using fixed CDNC avoids the uncertainties involved in the representation of aerosol processes in numerical models (Rothenberg et al., 2018), however, it limits potential feedbacks between clouds and aerosols, such as through aerosol scavenging (Yamaguchi et al., 2017)."

4. In addition, the 2M scheme does not suppress rain in high CCN concentrations to the extent that occurs in reality, where rain is suppressed pretty much when cloud drop effective radius is smaller than 14 micrometer (Chen et al., 2008; Freud et al., 2012; Gerber, 1996; Prabha et al., 2011; Rosenfeld et al., 2012; Van Zanten et al., 2005).

Reply: No model is perfect. Bin microphysical schemes have their own problems in predicting rain such as artificial broadening of the droplet size distribution due to numerical diffusion (Morrison et al., 2018) and a too early initialization of rain due to that. In addition, different bin microphysical schemes do not converge on the warm rain production rates (VanZanten et

al., 2011; Adrian Hill, personal communication). As was cited above, the limitations of our simulations are now better explained in the manuscript.

**5. The model misses the processes which can lead to positive net TOA warming due to aerosols, as simulated using SBM by Fan et al. (2012).**

Reply: Thank you. In a recent study (Dagan et al., 2019), using the same model used here, we showed that aerosol could generate net atmospheric column warming (up to 10 Wm-2) due to a reduction in outgoing longwave flux at TOA. This trend is driven by an increase in ice content and cover at the upper troposphere, a similar mechanism as described in Fan et al. (2012). Whether or not this reduction in longwave flux at TOA caused a positive net TOA warming (overcome the shortwave effect) is still an open question and depend on the environmental conditions (Koren et al., 2010). However, from figure 10 in the main text we can see that this does happen using our model on some of the summer days. This is already mentioned in the text:

"However, during the summer month, the mean effect is less negative  $(-1.7 \text{ W/m}^2)$  and for some of the days it could even be positive due to the effect of the CDNC on the ice water content (Fig. 5 and Table 2)."

In addition, the paper the reviewer pointed at (Fan et al., 2012) was added to the revised manuscript:

"It was shown that the total column atmospheric radiative warming  $(Q_R = (F_{SW}^{TOA} - F_{SW}^{SFC}) + (F_{LW}^{TOA} - F_{LW}^{SFC})$ , defined as the rate of net atmospheric diabatic warming due to radiative shortwave (SW) and longwave (LW) fluxes at the surface (SFC) and top of the atmosphere (TOA), when all fluxes positive downwards), is substantially increased with CDNC in a deep-cloud dominated case (by ~10 W/m2), while a much smaller increase (~1.6 W/m2) is shown in a shallow-cloud dominated case. This trend is caused by an increase in the upward mass flux of ice and water vapor to the upper troposphere that leads to reduced outgoing longwave radiation (Fan et al., 2012)."

**6. The model resolution of 1200 m is insufficient to resolve properly the trade wind cumulus.**

Reply: As was stated above, in order to sample a large range of initial conditions and cloud regimes we had to make some simulations choices. However, we still believe that our set-up captures the main processes acting on cloud-aerosol interaction (Naumann and Kiemle, 2019). We also compare our model results with ground-base measurements (Figs. S1 and S2, SI),

which demonstrates that the model does a reasonable job (see also a resent published research examining the effect of model resolution, using the same model and the same region, by Naumann and Kiemle (2019), demonstrating a "good skill" by the model compared to observations at a similar resolution).

The limitation of the horizontal resolution is now also mentioned in the revised manuscript: "We also note that using 1200 m horizontal resolution does not properly resolve all shallow cumulus clouds (Naumann and Kiemle, 2019)."

7. The simulation is allowed 12 hours for spin up time. This also means that clouds in air mass that enter the border of the domain (typically from the east) require that much time to spin up. But the whole domain of 3 degrees from east to west is 325 km, divided by 12 hours equals 27 km/hour or 7.5 m/s. This means that when air mass speed is larger than that, the spin up would not be reached throughout the domain. The actual mean surface wind at Barbados airport is easterly 14 knots, which is 7.2 m/s, with little variation between winter and summer. Therefore, most of the simulated clouds are well within the spin-up time.

Reply: Thank you. The 12 hours spin-up time which is mentioned in the manuscript is the spinup time required when starting the simulations from (ECMWF) reanalysis data. In our case we start our simulations from ICON simulations with a similar resolution (Klocke et al., 2017). The only main difference between the simulations of Klocke et al, (2017) and our simulations is the microphysical scheme. Hence, the spin-up time required is only few 10's of minutes (occurs within the cloud lifetime). An evidence for that can be seen in Figs. 8 and 16 in Dagan et al. (2019), which present time series of the domain mean properties and show that its responses to the microphysical perturbation within the first 30 min.

This is now better explained in the revised manuscript:

"Each simulation is conducted for 24 hours, starting from 12 UTC - 12 hours after the original simulations of Klocke et al., 2017 were initialized from reanalysis data, to reduce spin-up effects. Using initial and boundary conditions based on ICON simulations with similar resolution, as in Klocke et al. (2017), reduces the spin-up effects."

All these problems lend very little credibility to the conclusions with respect to the quantitative aerosol effects on the clouds. Presently SBM simulations are possible for the domain of this study, although quite more expensive. The fact that bulk models run faster is no longer a justification to use them for evaluating aerosol microphysical effects without addressing these issues.

Reply: As was mentioned above, using SBM is not possible for simulating 248 day-long, 300km by 300km domain simulations, which are needed for examining the dependency of the aerosol effect on the initial conditions in a robust manner. In addition, we note that there exists no consensus about the general preferability of SBM over bulk schemes based on observational evidence. We disagree with the implicit, yet unproven, assumption that only simulations based on SBM are credible, which would invalidate a large part of the existing literature.

In summary, I would recommend publication only after all these caveats will be explicitly highlighted, and the conclusions of the paper will clearly take them into account. What is the point to run very fast with 2-Moment bulk scheme when running in the wrong

**direction?**

Reply: As was mentioned above, it is far from being a consensus in our field that two-moment bulk schemes are running in the wrong direction – but that is not the focus of the presented work. However, we are aware of the limitations of cloud microphysics schemes and have highlighted this in the revised manuscript and the conclusions.

**Reviewer #2:**

This study is a follow-up to a paper exploring two case studies from the NARVAL campaign that uses ensemble simulations of two months (of which the original cases were a subset) to analyze the robustness of inferences regarding aerosol-cloud interactions that can be made on the basis of a small number of cases. Certain changes (such as in shortwave reflections and boundary layer deepening that lowers lower tropospheric stability) appear robust whereas others (such as cloud fraction and precipitation changes) appear less so. Seasonal differences in response can be explained via different responses in different cloud regimes, particularly due to ice-phase effects in deep clouds during the summer.

The manuscript is in very good shape and only requires some very minor revisions, in my estimation. If not for the comment below regarding the reasonableness of the "lower bound" language, I'd be happy to accept as is.

Reply: We would like to thank the reviser again for the constructive review and we are happy by the assessment that our paper is in very good shape.

Specific comments:

Page 2, Line 45: "As the anthropogenic activity. . ." is phrased somewhat awkwardly. Perhaps you can simplify to something like "Anthropogenic aerosol emissions may thus perturb Earth's radiation budget both directly by scattering and absorbing light and also indirectly through these cloud-mediated mechanisms."

Reply: Thank you for this suggestion. This sentence was changed according to the reviewer suggestion.

Page 4, Line 116: I'm glad you address this point. However, did you mean "interactions" or "feedbacks" instead of "involve"? Also, a relevant citation for the aerosol scavenging idea: Yamaguchi, T., Feingold, G., & Kazil, J. (2017). Stratocumulus to Cumulus Transition by Drizzle. Journal of Advances in Modeling Earth Systems, 9(6), 2333-2349. doi:10.1002/2017MS001104

Reply: Thank you. This sentence was changed and the suggested reference was added:

"Using fixed CDNC avoids the uncertainties involved in the representation of aerosol processes in numerical models (Rothenberg et al., 2018), however, it limits potential feedbacks between clouds and aerosols, such as through aerosol scavenging (Yamaguchi et al., 2017)."

Page 5, Line 138: I'm not convinced this is a reasonable lower bound, given that the relatively small domain size with fixed boundary conditions (which you argue would lead to an underestimate of aerosol effects) is not the only potential source of error, or necessarily the largest. I'd either like to see a fuller explanation of why the estimates should be seen as true lower bounds or a weaker statement simply explaining this particular source of error would tend to underestimate the effect compared to a simulation with a larger domain.

Reply: Thank you for this comment that helped us clarify this point. We agree with the reviewer that this statement was too strong in the previous version and hence in the revised manuscript we removed it:

"We note that although a 3 ° x 3° domain is larger than the domains used in many previous studies, it is still possible that the use of fixed boundary conditions for the different simulations under different CDNC conditions reduces some of the sensitivity as compared to simulations with larger domains such as in Dagan et al. (2019) (22 ° x 11°)."

Page 6, Figure 2: It would be helpful if "LP" were defined somewhere in the text in addition to in the figure captions.

**Reply: added.**

"This demonstrates that the lower tropospheric stability (LTS), top of atmosphere shortwave flux ( $\mathbf{F}_{SW}^{TOA}$ ), and the atmospheric column radiative term ( $Q_R$ ) are different in a statistically significant manner (p-value < 0.05) between the two different months. The differences in other parameters (cloud fraction – CF, liquid water path - LWP, ice water path – IWP, latent heat of precipitation – LP, and top of atmosphere longwave flux -  $\mathbf{F}_{LW}^{TOA}$ ) are not statistically significant (Table 1)."

Page 9, Line 226: I would add "likely" between "would" and "further" given that ice-phase microphysical changes can be quite complex.

**Reply: added.**

"Accounting for this effect would likely further increase the relative role of the Twomey effect compared to the cloud adjustment effects (CF and LWP/IWP adjustments)."

**Page 11, Line 275: How significant is 12 versus 8 in this context? Is there any way to quantify the variability we could expect in deep-cloud days due to chance?**

Reply: Thank you for this comment. It is indeed hard to say whether or not a 50% increase in the occurrence of deep convection between the two months (12 versus 8 days) is significant based only on this data. However, the difference between the summer and winter in the occurrence of deep convection in the Barbados region is well known (Stevens et al., 2016). In addition, in our data we do see a significant difference in the LTS between the two months (Fig. 2 and table 1 in the main text), which is consistent with the increase in deep convection occurrence.

Based on this comment we have added clarification to the revised manuscript:

"The separation into different cloud regimes also demonstrates that more deep-cloud days are occurring during the summer month as compared to the winter month (12 compare to 8) and that the deep clouds during summer are deeper and contain more water. The larger occurrence of deep convection during the summer month is consistent with the statistically significant reduction in LTS (Fig. 2 and Table 1) and is expected based on the local seasonality (Stevens et al., 2016)."

References: There are some typos and weird formatting issues with some references. A quick proofread should sort most of those out.

Reply: Thank you. The manuscript (including the references section) went thought proofreading.

**References**

Dagan, G., Koren, I., and Altaratz, O.: Competition between core and periphery-based processes in warm convective clouds–from invigoration to suppression, Atmospheric Chemistry and Physics, 15, 2749-2760, 2015a.

Dagan, G., Koren, I., and Altaratz, O.: Aerosol effects on the timing of warm rain processes, Geophysical Research Letters, 42, 4590-4598, 10.1002/2015GL063839, 2015b.

Dagan, G., Koren, I., Altaratz, O., and Heiblum, R. H.: Aerosol effect on the evolution of the thermodynamic properties of warm convective cloud fields, Scientific Reports, 6, 38769, 10.1038/srep38769

https://www.nature.com/articles/srep38769#supplementary-information, 2016.

Dagan, G., Koren, I., Altaratz, O., and Heiblum, R. H.: Time-dependent, non-monotonic response of warm convective cloud fields to changes in aerosol loading, Atmos. Chem. Phys., 17, 7435-7444, 10.5194/acp-17-7435-2017, 2017.

Dagan, G., Koren, I., and Altaratz, O.: Quantifying the effect of aerosol on vertical velocity and effective terminal velocity in warm convective clouds, Atmospheric Chemistry and Physics, 18, 6761-6769, 2018.

Dagan, G., Stier, P., Christensen, M., Cioni, G., Klocke, D., and Seifert, A.: Atmospheric energy budget response to idealized aerosol perturbation in tropical cloud systems, Atmos. Chem. Phys. Discuss., https://doi.org/10.5194/acp-2019-813, in press, 2019.

Fan, J., Leung, L. R., Rosenfeld, D., Chen, Q., Li, Z., Zhang, J., and Yan, H.: Microphysical effects determine macrophysical response for aerosol impacts on deep convective clouds, Proceedings of the National Academy of Sciences, 110, E4581-E4590, 2013.

Grabowski, W. W., Morrison, H., Shima, S.-I., Abade, G. C., Dziekan, P., and Pawlowska, H.: Modeling of cloud microphysics: Can we do better?, Bulletin of the American Meteorological Society, 100, 655-672, 2019.

Heiblum, R. H., Altaratz, O., Koren, I., Feingold, G., Kostinski, A. B., Khain, A. P., Ovchinnikov, M., Fredj, E., Dagan, G., and Pinto, L.: Characterization of cumulus cloud fields using trajectories in the center of gravity versus water mass phase space: 2. Aerosol effects on warm convective clouds, Journal of Geophysical Research: Atmospheres, 2016.

Khain AP, Beheng KD, Heymsfield A, Korolev A, Krichak SO, Levin Z, Pinsky M, Phillips V, Prabhakaran T, Teller A, van den Heever SC. Representation of microphysical processes in

cloud-resolving models: Spectral (bin) microphysics versus bulk parameterization. Reviews of Geophysics. 2015 Jun;53(2):247-322.

Klocke, D., Brueck, M., Hohenegger, C., and Stevens, B.: Rediscovery of the doldrums in storm-resolving simulations over the tropical Atlantic, Nature Geoscience, 10, 891, 2017.

Koren, I., Remer, L. A., Altaratz, O., Martins, J. V., and Davidi, A.: Aerosol-induced changes of convective cloud anvils produce strong climate warming, Atmospheric Chemistry and Physics, 10, 5001-5010, 10.5194/acp-10-5001-2010, 2010.

Koren, I., and Feingold, G.: Aerosol-cloud-precipitation system as a predator-prey problem, Proceedings of the National Academy of Sciences of the United States of America, 108, 12227-12232, 10.1073/pnas.1101777108, 2011.

Morrison, H., Witte, M., Bryan, G. H., Harrington, J. Y., and Lebo, Z. J.: Broadening of modeled cloud droplet spectra using bin microphysics in an Eulerian spatial domain, Journal of the Atmospheric Sciences, 75, 4005-4030, 2018.

Naumann AK, Kiemle C. The vertical structure and spatial variability of lower tropospheric water vapor and clouds in the trades. Atmospheric Chemistry and Physics, https://doi.org/10.5194/acp-2019-1015 2019.

Pinsky, M., Mazin, I., Korolev, A., and Khain, A.: Supersaturation and diffusional droplet growth in liquid clouds, Journal of the Atmospheric Sciences, 70, 2778-2793, 2013.

Stevens, B., Farrell, D., Hirsch, L., Jansen, F., Nuijens, L., Serikov, I., Brügmann, B., Forde, M., Linne, H., and Lonitz, K.: The Barbados Cloud Observatory: Anchoring investigations of clouds and circulation on the edge of the ITCZ, Bulletin of the American Meteorological Society, 97, 787-801, 2016.

VanZanten, M. C., Stevens, B., Nuijens, L., Siebesma, A. P., Ackerman, A., Burnet, F., Cheng, A., Couvreux, F., Jiang, H., and Khairoutdinov, M.: Controls on precipitation and cloudiness in simulations of trade-wind cumulus as observed during RICO, Journal of Advances in Modeling Earth Systems, 3, 2011.

Xue, L., Fan, J., Lebo, Z. J., Wu, W., Morrison, H., Grabowski, W. W., Chu, X., Geresdi, I., North, K., and Stenz, R.: Idealized Simulations of a Squall Line from the MC3E Field Campaign Applying Three Bin Microphysics Schemes: Dynamic and Thermodynamic Structure, Monthly Weather Review, 145, 4789-4812, 2017.